# Endothelial Cell Targeting by cRGD-Functionalized Polymeric Nanoparticles under Static and Flow Conditions

**DOI:** 10.3390/nano10071353

**Published:** 2020-07-10

**Authors:** Lucía Martínez-Jothar, Arjan D. Barendrecht, Anko M. de Graaff, Sabrina Oliveira, Cornelus F. van Nostrum, Raymond M. Schiffelers, Wim E. Hennink, Marcel H. A. M. Fens

**Affiliations:** 1Department of Pharmaceutics, Utrecht Institute for Pharmaceutical Sciences, Utrecht University, Universiteitsweg 99, 3584 CG Utrecht, The Netherlands; lucia.martinez.jothar@gmail.com (L.M.-J.); s.oliveira@uu.nl (S.O.); c.f.vannostrum@uu.nl (C.F.v.N.); w.e.hennink@uu.nl (W.E.H.); 2CDL Research, University Medical Center Utrecht, 3584 CX Utrecht, The Netherlands; a.barendrecht@umcutrecht.nl (A.D.B.); r.schiffelers@umcutrecht.nl (R.M.S.); 3Hubrecht Imaging Center, Hubrecht Institute, Royal Netherlands Academy of Arts and Sciences and University Medical Center Utrecht, Uppsalalaan 8, 3584 CT Utrecht, The Netherlands; a.graaff@hubrecht.eu; 4Division of Cell Biology, Neurobiology and Biophysics, Department of Biology, Utrecht University, Padualaan 8, 3584 CH Utrecht, The Netherlands

**Keywords:** nanoparticles, endothelial cells, RGD, targeting, physiological flow, PLGA

## Abstract

Since α_v_β_3_ integrin is a key component of angiogenesis in health and disease, Arg-Gly-Asp (RGD) peptide-functionalized nanocarriers have been investigated as vehicles for targeted delivery of drugs to the α_v_β_3_ integrin-overexpressing neovasculature of tumors. In this work, PEGylated nanoparticles (NPs) based on poly(lactic-co-glycolic acid) (PLGA) functionalized with cyclic-RGD (cRGD), were evaluated as nanocarriers for the targeting of angiogenic endothelium. For this purpose, NPs (~300 nm) functionalized with cRGD with different surface densities were prepared by maleimide-thiol chemistry and their interactions with human umbilical vein endothelial cells (HUVECs) were evaluated under different conditions using flow cytometry and microscopy. The cell association of cRGD-NPs under static conditions was time-, concentration- and cRGD density-dependent. The interactions between HUVECs and cRGD-NPs dispersed in cell culture medium under flow conditions were also time- and cRGD density-dependent. When washed red blood cells (RBCs) were added to the medium, a 3 to 8-fold increase in NPs association to HUVECs was observed. Moreover, experiments conducted under flow in the presence of RBC at physiologic hematocrit and shear rate, are a step forward in the prediction of in vivo cell–particle association. This approach has the potential to assist development and high-throughput screening of new endothelium-targeted nanocarriers.

## 1. Introduction

Angiogenesis, an essential process for tissue formation and repair, relies on complex cell–cell and cell–matrix interactions mediated by adhesion molecules [1,2,3]. Integrins are transmembrane receptors consisting of an α and a β subunit that comprise one of the most extensive families of cell adhesion molecules. The extracellular domain of integrins binds to ligands present on proteins of the extracellular matrix (ECM) or to receptors on neighboring cells, while the cytoplasmic domain transfers external stimuli to the cytoskeleton and plays a role in activation of intracellular signaling pathways [4]. Integrin α_v_β_3_, one of the 24 integrin subtypes, is highly and luminally expressed by activated (angiogenic) endothelial cells but not on quiescent cells, and is therefore an important mediator of angiogenesis under normal and pathological conditions [5,6]. The most important ligands for α_v_β_3_ integrins, are peptide sequences with an arginine-glycine-aspartic acid (RGD) motif, present in proteins such as lactadherin, fibronectin, vitronectin, laminin and von Willebrand factor [7,8]. In addition to participating in the regulation of cell adhesion, migration and growth during tumor angiogenesis, integrin α_v_β_3_ is also associated with increased proliferation, survival and/or metastasis of some cancers [9,10,11], which positions it as an important pharmacological target for treatment [12]. While the inhibition of integrin α_v_β_3_ suppresses angiogenesis, tumor growth and metastasis in pre-clinical studies [13,14], these findings have not yet been successfully translated to the clinic [15]. Importantly, some researchers have explored the possibility of using integrin α_v_β_3_ as a target for the delivery of anti-cancer agents via RGD-functionalized nanocarriers [16,17,18,19,20,21]. RGD-functionalized polymeric nanoparticles (NPs) loaded with cytotoxic drugs have shown enhanced anti-cancer activity both in vitro and in vivo compared to non-functionalized polymeric NPs and/or free drugs [22,23,24,25]. The better efficacy of RGD-NPs in vivo can be attributed to several factors, including improved pharmacokinetics of the drugs by employing PEGylated NPs (i.e., prolonged circulation time), as well as the interactions between RGD grafted on the surface of NPs and α_v_β_3_ integrins present on angiogenic endothelial cells. In respect to this, the number and density of RGD moieties on the surface of NPs is very likely to play a role on the extent of target cell binding. In fact, the relevance of multivalent interactions has been highlighted by the superior binding affinity and internalization of multimeric-RGD structures (i.e., radiolabeled or fluorescent peptides) in comparison to their monomeric analogues [26,27,28,29]. These findings prompted us to study the influence of ligand density on the interaction between NPs functionalized with cRGD and human umbilical vein endothelial cells (HUVECs), an extensively used in vitro model for angiogenic endothelial cells [30,31]. For this purpose, HUVECs were incubated with poly(lactic-co-glycolic acid) (PLGA)-based NPs functionalized with cRGD at different densities, and nanoparticle-cell interactions were evaluated under static incubation conditions. Since α_v_β_3_ integrins are preferentially expressed by activated endothelial cells of nascent blood vessels, we also studied association under physiological flow conditions. For this purpose, we used a perfusion system and conducted experiments using either merely cell culture medium or a suspension of washed red blood cells (RBCs) at physiological hematocrit as flow medium. It has been implied that culturing cells and having test conditions under flow provide a better representation of the (patho)physiological conditions under which nanocarrier-endothelial cell contact occurs in vivo [32,33]. In this regard, the generation of flow-induced shear stress can influence cellular responses such as cell–cell adhesion [34], NP uptake [35,36,37,38] and cytocompatibility with lipid, polymeric and metallic NPs [39,40].

Taking this into consideration, we conducted studies with the purpose to gain a better understanding of the impact of (a) the ligand density of NPs and (b) the test conditions (i.e., static vs. flow, and medium with and without washed RBCs), on the cell binding and uptake of α_v_β_3_ targeted and non-targeted PLGA-NPs by HUVECs.

## 2. Materials and Methods

### 2.1. Chemicals

PLGA (50:50 ratio DL-lactide/glycolide, IV 0.39 dl/g, Mw ~44,000 Da) was obtained from Corbion (Gorinchem, the Netherlands). Poly(lactide-*co*-glycolide)-*b*-poly(ethylene glycol)-maleimide (maleimide-PEG_5,000_-PLGA_20,000_) was purchased from Polyscitech, Akina Inc (West Lafayette, IN, USA). Poly(vinyl alcohol) (PVA) of Mw 30,000–70,000 Da, 87–90% hydrolyzed, L-cysteine hydrochloride monohydrate and hydroxylamine hydrochloride were acquired from Sigma-Aldrich (Steinheim, Germany). Dicyclohexylcarbodiimide (DCC) was purchased from Aldrich (Steinheim, Germany). 4-(Dimethylamino)pyridine was obtained from Fluka (Steinheim, Germany). Dichloromethane (DCM), diethyl ether, acetonitrile and dimethyl sulfoxide (DMSO) were obtained from Biosolve (Valkenswaard, the Netherlands). Cyanine5 free carboxylic acid (Cy5-COOH) was purchased from Lumiprobe (Hannover, Germany). The cyclic peptide c[RGDfK(Ac-SCH2CO)] (Mw: 719.8 Da) was purchased from Peptides International (Louisville, KY, USA). HEPES, EDTA disodium salt dihydrate, gelatin from bovine skin and Dulbecco’s phosphate buffered saline (8.0 g NaCl, 1.15 g Na_2_HPO_4_, 0.2 g KCl and 0.2 g KH_2_PO_4_ in 1 L of water, pH 7.4) were products of Sigma (Steinheim, Germany). Micro BCA^TM^ Protein Assay Kit was purchased from Thermo Scientific (Illinois, USA).

### 2.2. Synthesis of Cy5 Labeled Poly(D,L-lactic-co-glycolic-co-hydroxymethyl glycolic acid) (Cy5-PLGHMGA)

Poly(D,L-lactic-co-glycolic-co-hydroxymethyl glycolic acid) (PLGHMGA) was synthesized by ring-opening polymerization of D,L-lactide and benzyloxymethyl glycolide at a molar ratio of 65:35, as reported previously [41,42]. The fluorescent label Cy5-COOH was covalently attached to the pendant hydroxyl groups of PLGHMGA by DCC-DMAP chemistry. Briefly, 50 mg of PLGHMGA (M_n_ 43 kDa) was dissolved in 5 mL of dry DCM and placed under stirring in a round bottom flask with a rubber stopper. Subsequently, 18 µL of DCC (1.6 mg/mL dissolved in DCM, 140 nmol), 60 µL of Cy5-COOH (1 mg/mL dissolved in DCM, 116 nmol) and 4 µL of DMAP (0.5 mg/mL DCM, 16 nmol) were added to the polymer solution and the mixture was stirred and flushed with nitrogen for 10 min at room temperature (RT). The reaction was allowed to proceed overnight under stirring and protected from light. Next, the reaction mixture was concentrated under reduced pressure (removal of ~80% of the DCM volume) and the polymer was precipitated in cold diethylether (25 mL). The fluorescent polymer was recovered by filtration, dried under reduced pressure, dissolved in 2 mL of DCM and dialyzed against DMSO for 48 h to remove the free dye (dialysis membrane MWCO 6–8 kDa, Spectra/Por, Spectrum Labs, Piraeus, Greece). The purified polymer was freeze dried at −40 °C, <1 mbar (Christ Alpha 1-2 freeze dryer). The Cy5 labeled polymer (yield 92%) was characterized by gel permeation chromatography (GPC) using two PL-gel 5 µm Mixed-D columns and tetrahydrofuran/lithium chloride 10 mM as the mobile phase (1 mL/min) at 60 °C. Dual RI and UV detection (λ 650 nm) was used for analysis, and polystyrene standards (EasiCal Agilent, California, USA) were used for calibration. The average molecular weight (M_w_) of the polymer was 62 kDa (number average molecular weight (M_n_) was 42 kDa, PDI 1.46) and no signal for the free dye was detected after purification.

### 2.3. Preparation and Characterization of the Polymeric Nanoparticles (NPs)

Fluorescent NPs were prepared by double emulsion solvent evaporation method because they are intended for future use as carriers of hydrophilic therapeutic agents [43,44]. A mixture of PLGA, maleimide-PEG-PLGA and Cy5-PLGHMGA (ratio 65/20/15 wt%) was dissolved in DCM at 5% *w/v*. To prepare the first W/O (water-in-oil) emulsion, 100 µL of water was added to 1 mL of polymer solution and the mixture was emulsified using a probe sonicator (SONOPULS HD 2200 Bandelin, Berlin, Germany) for 1 min at 20 W power in an ice bath. The W/O emulsion was subsequently added dropwise to 10 mL of an aqueous solution of PVA 5% *w/w*. The addition was done under sonication for 2 min at 20 W in an ice bath and resulted in the formation of a W/O/W (water-in-oil-in-water) emulsion. This emulsion was stirred (600 rpm) for 2 h at RT to evaporate DCM. The NPs formed after solvent evaporation were collected by centrifugation for 20 min, 20,000× *g* at 4 °C, washed twice with HEPES 10 mM (pH 7.0) and once with distilled water. After the last washing, the NPs were resuspended in 1 mL of distilled water and divided into aliquots of 250 µL. One of the aliquots was freeze dried in order to determine the yield of the preparation process, while the other aliquots were supplemented with sucrose at a final concentration of 5% *w/v* prior to freeze drying (−40 °C, <1 mbar, Christ Alpha 1-2 freeze dryer).

The size of the NPs was determined by Dynamic Light Scattering (Zetasizer Nano S, Malvern, Worcestershire, UK) at 25 °C in MilliQ water and their zeta potential (Zetasizer Nano Z, Malvern) was determined at 25 °C in HEPES 10 mM, pH 7.0.

### 2.4. Conjugation of cRGD to the NPs

The cRGD peptide was conjugated to the fluorescent NPs by maleimide-thiol chemistry as described previously [45]. Briefly, c[RGDfK(Ac-SCH2-CO)] was deprotected by incubation for 30 min at RT in a buffer containing 10 mM HEPES/0.4 mM of EDTA/45 mM hydroxylamine (pH 7.0), in order to remove the acetyl group to generate a free thiol per peptide molecule. Next, deprotected cRGD was conjugated to the fluorescent NPs at different molar ratios cRGD to maleimide-polymer, namely 1:10 (“low” cRGD-NPs), 1:5 (“medium” cRGD-NPs) and 1:2 (“high” cRGD-NPs), as follows. Freeze dried NPs were resuspended in distilled water and recovered by centrifugation at 3000× *g*, 10 min, 4 °C. The pelleted NPs were resuspended in a buffer of 10 mM HEPES/0.4 mM EDTA (pH 7.0) to a concentration of ~7 mg/mL and aliquots containing 2 mg of NPs were incubated with deprotected cRGD (13, 5.0 and 2.5 µL of cRGD 0.45 mg/mL to 300 µL of NPs suspension), for 30 min at RT in the dark. The NPs were recovered by centrifugation (3000× *g*, 10 min, 4 °C), the supernatant was removed and stored for analysis of the cRGD concentration (described in Section 2.5) and the pellet was resuspended in 10 mM HEPES/0.4 mM EDTA to a concentration of ~7 mg/mL. Cysteine dissolved in 10 mM HEPES/0.4 mM (pH 7.0) was added to the NPs suspension in a molar ratio of 2:1 cysteine to maleimide-polymer (6.0 µL of cysteine 0.5 mg/mL to 300 µL of NPs suspension) to block the remaining unreacted maleimide groups. Subsequently, the suspension was incubated for 30 min at RT in the dark, the NPs were recovered by centrifugation (3000× *g*, 10 min, 4 °C), the supernatant was stored for quantification of cysteine (Section 2.5) and the pellet was washed with 10 mM HEPES/0.4 mM EDTA. Finally, the NPs pellet was resuspended in PBS at a concentration of 10 mg/mL and stored at 4 °C in the dark until further use.

Control (non-targeted) NPs functionalized with cysteine at a molar ratio of 2:1 cysteine to maleimide-polymer (Cys-NPs) were prepared as described for the NPs functionalized with cRGD, but skipping the addition of this peptide.

The size and zeta potential of the cRGD-NPs and the Cys-NPs were determined as described in Section 2.3.

### 2.5. Quantification of cRGD by HPLC to Determine the Conjugation Efficiency to the NPs

The concentration of cRGD present in the supernatants of the pelleted functionalized NPs was quantified by HPLC (Waters Alliance System) equipped with a UV detector (analysis at 214 nm) as previously reported [45]. Briefly, the samples (50 µL injection volume) were run through a XBridge BEH C18 column (3.5 µm, 2.1 mm × 100 mm) using a gradient from 100% eluent A (94.9% H_2_O, 5.0% ACN, 0.1% acetic acid) to eluent B (94.9% ACN, 5.0% H_2_O, 0.1% acetic acid) in 12 min followed by 100% eluent B in 1 min. The detection limit was 2 µg/mL.

The cysteine concentration in the supernatants of the pelleted NP dispersions was quantified by Micro BCA assay according to the protocol of the manufacturer. Detection limit was 2 µg/mL.

The conjugation efficiency was calculated as:
(1)$$Conjugation\;efficiency\;\left(\%\right)=\left(1-\left(\frac{\left[Ligand\;in\;the\;supernatant\right]}{\left[Ligand\;added\;for\;the\;conjugation\;reaction\right]}\right)\right)\times 100\%$$

Since cysteine was added in excess compared to the maleimide groups in the NPs (2:1 molar ratio), the maximum achievable conjugation efficiency was 50%. Therefore, the conjugation efficiency of cysteine was normalized to this value.

### 2.6. Cell Culture Conditions

Pooled human umbilical vein endothelial cells (HUVECs) were obtained from Lonza (Verviers, Belgium) and cultured in Endothelial Basal Medium (EBM-2) supplemented with growth factors (Growth Medium 2 SupplementMix^®^, PromoCell, Heidelberg, Germany) and antibiotics (gentamicin/amphotericin B, Gibco, New York, USA) up to passage number 6. The cells were incubated at 37 °C and 5% CO_2_, in a humidified atmosphere during culture as well as during the experiments (Section 2.7, Section 2.8 and Section 2.9).

For the flow experiments, HUVECs were seeded on a perfusion slide (µ-Slide I^0.6^ Luer^®^, Ibidi, Martinsried, Germany) at a density of 2.4 × 10^5^ cells/slide and allowed to attach for 2 h. Next, the slide was connected to a pump-controlled Perfusion Set yellow/green (50 cm, ID 1.6 mm, Ibidi, Martinsried, Germany) and the cells were cultured in EBM-2 medium supplemented with growth factors and antibiotics (EGM-2), under continuous unidirectional flow (shear rate 300 s^−1^, shear stress 0.3 N/m^2^ (3.0 dyn/cm^2^), viscosity 1 mPa.s (0.01 dyn.s/cm^2^) using PumpControl v1.5.2 software) at 37 °C and 5% CO_2_.

### 2.7. Association of Cys-NPs and cRGD-NPs with HUVECs under Static Conditions

HUVECs were seeded in a 12 well plate at a density of 50,000 cells/well. After 24 h of incubation, the cell medium was refreshed and fluorescent Cys-NPs or fluorescent cRGD-NPs dispersed in PBS were added to the cells at final concentrations of 0.016, 0.08 and 0.4 mg/mL. HUVECs were incubated with the NPs for 1 or 3 h, after which they were washed twice with PBS and detached from the wells using trypsin/EDTA (0.05%). Trypsin was neutralized with Dulbecco’s PBS containing 0.5% bovine serum albumin, the cells were recovered by centrifugation at 300× *g* for 5 min at RT and the supernatant was removed. The cell pellet was resuspended in PBS also containing 0.5% bovine serum albumin. The fluorescence associated to the cells was determined by flow cytometry (BD FACSCanto II, BD Biosciences) using an APC laser (λ 660 nm, used to detect the Cy5 signal from the NPs). Initially, HUVECs were gated by plotting FSC/SSC and 10,000 events were recorded (gate P1). The mean fluorescence intensity (MFI) was determined for the total cell population (P1) and subsequent gating of P1 was done to calculate the percentage of cells that showed above background fluorescence (gate P2), using untreated HUVECs as a control.

### 2.8. Uptake of Cys-NPs and cRGD-NPs by HUVECs under Static Incubation Conditions

Lab-Tek 16 well chamber slides (Nunc^®^) were coated with 0.5% gelatin from bovine skin (30 min, 37 °C) followed by 0.5% glutaraldehyde in PBS (10 min, RT) and wells were finally washed three times with PBS. HUVECs were seeded in the coated wells at a density of 10,000 cells/well and incubated overnight at 37 °C. Next, the cell medium was refreshed and fluorescent Cys-NPs or fluorescent cRGD-NPs dispersed in PBS were added to the cells at a final concentration of 0.4 mg/mL. The cells were incubated with the NPs for 1 or 3 h, after which they were washed twice with PBS and fixed with 2% paraformaldehyde/0.2% glutaraldehyde in PBS for 1 h at RT and then stored overnight at 4 °C. The nuclei were stained using Hoechst 33342 (Fluka), 1 µg/mL in PBS for 20 min, washed once with PBS and the F-actin cytoskeleton was stained with phalloidin Alexa Fluor 488 (Life Technologies, Carlsbad, California, USA), 1:50 in PBS for 30 min. After washing, the cells were mounted with FluorSave^®^ reagent (Calbiochem, San Diego, California, USA). HUVECs were visualized by confocal microscopy using a Leica TCS SP8 X confocal microscope with a white light laser (continuous spectral output between λ 470–670 nm) and a 63×/1.20 water immersion objective. Two fields were imaged per condition and images were captured in three channels: Hoechst 33342 for nuclei (λ_ex_ 419 nm, λ_em_ 500 nm), phalloidin 488 (λ_ex_ 510 nm, λ_em_ 562 nm) for F-actin cytoskeleton and Cy5 to for visualization of the NPs (λ_ex_ 660 nm, λ_em_ 700 nm).

### 2.9. Association of Cys-NPs and RGD-NPs with HUVECs under Flow

The experiments under flow were performed using an Ibidi pump system^®^ equipped with PumpControl v1.5.2 software. The cells were cultured under unidirectional flow as mentioned in Section 2.6 for 3 days prior to the addition of NPs.

#### 2.9.1. Experiments under Flow: 1 h Incubation in EBM-2 Medium

The medium in the perfusion system was replaced with EBM-2 medium and the system was allowed to equilibrate for 2 min under flow. Next, Cys-NPs or cRGD-NPs dispersed in PBS were added to the system (flow was stopped during addition) at a final concentration of 0.08 mg/mL and the HUVECs were incubated with the NPs for 1 h at 37 °C under flow (shear rate 300 s^−1^, flow rate 5.0 mL/min, viscosity 1 mPa.s, shear stress 0.3 N/m^2^). This led to a calculated overall exposure of 24 mg NPs to HUVECs. The incubation was followed by medium refreshment and washing of the cells with a flow of EBM-2 for 5 min. Finally, the slide was disconnected from the system and an additional washing of the cells was done manually by carefully flushing 3 mL of EBM-2 through the slide. The cells were then fixed with 2% paraformaldehyde/0.2% glutaraldehyde in PBS for 1 h at RT, and then stored overnight at 4 °C. After fixation, HUVECs were stained with Hoechst 33342 and phalloidin 488 as described in Section 2.8. Cell images (epifluorescence) were captured using a Zeiss Axio Observer Z1 microscope equipped with Colibri LEDs (Carl Zeiss filter set 49 for HOECHST 33342, filter set 10 for Phalloidin 488 and filter set 50 for Cy5), using a 40× objective (Fluar 40×/1.30 Oil) and an AxioCam MRm. Ten images, captured along the slide, were processed using ImageJ image analysis software (NIH, USA) as follows: first, background noise was eliminated for all images. Then, A threshold was set for the images from the experiments under flow in EBM-2 (1 h and 16 h) and in EBM-2 with washed RBCs (16 h) in order to reduce remaining background signal. Finally, the Integrated Density was determined in the channel containing the signal of the Cy5 labeled NPs. For the figure panels, images were cropped to show a representative area for each condition.

#### 2.9.2. Experiments under Flow: 16 h Incubation in EBM-2 Medium

The medium in the system was replaced by EBM-2 medium and the system was allowed to equilibrate for 2 min. Cys-NPs or cRGD-NPs dispersed in PBS were added at a final concentration of 0.08 mg/mL and the cells were incubated under flow (shear rate 300 s^−1^, flow rate 5.0 mL/min., viscosity 1 mPa.s, shear stress 0.3 N/m^2^) for 16 h at 37 °C and 5% CO_2_. This led to a calculated overall exposure of 384 mg NPs to HUVECs. Subsequently, the medium containing NPs was replaced by plain EBM-2 medium and the cells were washed under flow for 5 min. Finally, the slide was disconnected from the system, an additional washing of the cells was done manually by flushing 3 mL of EBM-2 through the slide, and the cells were fixed and stained as described in Section 2.9.1.

#### 2.9.3. Experiments under Flow, 16 h Incubation in Washed Red Blood Cells (RBCs)

Whole human blood anticoagulated using sodium citrate 3.2% (from a single donor) was obtained from the mini donor service at University Medical Center Utrecht (the Netherlands) and centrifuged at 1000× *g* for 15 min. Next, RBCs were separated from the plasma and the buffy coat, and washed once with PBS. A suspension of washed RBCs was prepared in EBM-2 (20 mL washed RBC and 26 mL medium). The medium in the Ibidi system was replaced with the RBCs suspension (hematocrit 32% as measured by Cell-Dyn^®^ Hematology Analyzer, Abbott Laboratories, Abbott Park, Illinois, USA) and the system was allowed to equilibrate for 5 min. Cys-NPs or cRGD-NPs dispersed in PBS were added at a final concentration of 0.08 mg/mL and the cells were incubated under flow (shear rate 300 s^−1^, flow rate 5.0 mL/min., viscosity 1.8 mPa.s, shear stress 0.54 N/m^2^) for 16 h at 37 °C and 5% CO_2_. The washed RBCs suspension was replaced by EBM-2 medium and the HUVECs were washed for 5 min. The slide was disconnected from the system and an additional washing was done manually by flushing 3 mL of EBM-2 through the slide. The cells were subsequently fixed and stained as described in Section 2.9.1.

### 2.10. Statistical Analysis

The statistical differences between the experimental groups were analyzed by either one-way or two-way ANOVA using GraphPad Prism 8 software (Version 8.4.2, GraphPad Software Inc., La Jolla, CA, USA).

## 3. Results

### 3.1. Preparation and Characterization of the Polymeric Nanoparticles (NPs)

Cy5 labeled maleimide-PEG-PLGA NPs were prepared by double emulsion solvent evaporation method using a blend of PLGA, maleimide-PEG-PLGA and Cy5-PLGHMGA at 65/20/15 wt%, respectively. The yield of NPs preparation was 50%. For the preparation of the targeted NPs, different molar ratios of cRGD to maleimide-PEG-PLGA were used in order to obtain formulations with different surface ligand densities, designated as low (1:10 molar ratio), medium (1:5 molar ratio) and high (1:2 molar ratio), while NPs conjugated to cysteine were prepared as a non-targeted control. Cys-NPs and cRGD-NPs with different ligand densities were similar in size (~300–330 nm), surface area (0.31–0.35 µm^2^) and surface charge (Table 1). The conjugation efficiency of cysteine to the NPs was ~40% and the conjugation efficiencies of cRGD to the NPs ranged between 40–70%. These efficiencies resulted in the successful preparation of NPs with either cysteine or different cRGD surface densities (Table 1 and Appendix A). The ligand density on the high cRGD-NPs was ~3.5 times higher than on the medium cRGD-NPs and ~10 times higher than on the low cRGD-NPs. The particles had a slightly negative zeta potential between −10.1 and 12.2 mV in agreement with our previous study [45].

### 3.2. Association of Cys-NPs and RGD-NPs with HUVECs under Static Incubation Conditions

Dispersions of different concentrations of fluorescently labelled NPs with and without cRGD decoration, were incubated with HUVECs for 1 or 3 h and the association of NPs to the cells was assessed by flow cytometry (Figure 1 and Table 2). For high cRGD-NPs at 0.4 mg/mL (highest concentration tested), approximately 100% of the HUVECs were associated to NPs already after 1 h of incubation. Based on the average fluorescent signals obtained by flow cytometry, cell association of cRGD-NPs was 2.9 to 4.0-fold higher for high cRGD-NPs compared to low cRGD-NPs after 1 h of incubation, and 2.2 to 3.7-fold higher after 3 h of incubation (at different concentrations of NPs). Medium cRGD-NPs displayed values that were in between the low- and high-density NPs. An increase in the incubation time of all cRGD-NPs with HUVECs from 1 to 3 h resulted in 1.3 to 2.3 times more cell association. The increase in fluorescence intensity observed upon incubation of the cells with these NPs for 3 h, shows that more particles were associated per cell.

Confocal imaging of Cys-NPs showed negligible intracellular accumulation, which is in agreement with its poor cell association as demonstrated by flow cytometry (Figure 2). In contrast, cytoplasmic accumulation (confirmed by phalloidin staining of F-actin cytoskeleton) was observed upon incubation with the different cRGD-NPs. Similar to the flow cytometry results (Figure 1), confocal microscopy observations indicate that the cRGD density on the surface of the NPs is positively correlated with their accumulation (uptake) in HUVECs (Figure 2). Moreover, the signal of the cRGD-NPs in the cytoplasm showed a dotted pattern rather than diffused fluorescence, suggesting that the NPs localize to endosomes after their uptake [46,47,48].

### 3.3. Association of Cys-NPs and RGD-NPs with HUVECs under Flow

Incubation of HUVECs with Cys-NPs (0.08 mg/mL) dispersed in EBM-2 for 1 h under flow (at a share rate of 300 s^−1^, which is comparable to human carotid arteries) resulted in negligible association of these nanoparticles with HUVECs, as shown by epifluorescence microscopy images (Figure 3A). In contrast, a dotted pattern inside the cytoplasm of the cells was observed for HUVECs incubated with cRGD-NPs under the same conditions (Figure 3A). Based on the image analysis of the fluorescent signal from the NPs (expressed as integrated density), cell association of cRGD-NPs with HUVECs was dependent on ligand density (Figure 3B), with the association of high cRGD-NPs being 9-fold higher than the association of low cRGD-NPs.

Cell association of cRGD-NPs incubated with HUVECs in EBM-2 under flow for 16 h at 37 °C was also dependent on cRGD-density, as shown by epifluorescence microscopy visualization and image analysis (Figure 4). In this case, the association of high cRGD-NPs was 6.5-fold higher than that of low cRGD-NPs. For all cRGD-NPs tested, higher associations were observed after 16 h incubation compared to the 1 h incubation (6 to 8-fold difference, Figure 5A).

HUVECs were also incubated for 16 h under flow with Cys-NPs and cRGD-NPs (0.08 mg/mL) dispersed in EBM-2 containing washed RBCs (~30% hematocrit). The microscopic observations at the end of this experiment showed alterations in cell morphology (i.e., shrinking and in some cases rounding-up) compared to the cells from the flow experiments without RBCs. In some instances, the actin cytoskeleton seemed somewhat disrupted (based on the observations made with phalloidin staining). These morphological changes can likely be a result of the collisions between RBCs and HUVECs. In line with the previous experiments under static and flow conditions (EBM-2), cell association was observed for cRGD-NPs, but not for Cys-NPs (Figure 6A). Association of cRGD-NPs was dependent on ligand density and it was 2.3-fold higher for high cRGD-NPs than for low cRGD-NPs (Figure 6B).

Notably, the association of cRGD-NPs under flow was higher for all cRGD densities when the experiment was conducted in EBM-2 containing washed RBCs compared to EBM-2 without RBCs (Figure 5B). The presence of RBCs had a large influence on the cell association of low and medium cRGD-NPs, which increased by a factor 6 and 8, respectively. In comparison, the association of high cRGD-NPs increased by a factor 3.

## 4. Discussion

NPs were prepared for specific targeting of α_v_β_3_ integrins by conjugation of cRGD peptides (bearing one thiol group per cRGD molecule) to the surface of the Cy5 labeled maleimide-PEG-PLGA NPs, by maleimide-thiol chemistry. As non-targeted control NPs, cysteine was conjugated. Particle sizes varied between 300–330 nm with a PDI ~0.2 and a slightly negative zeta potential. The conjugation efficiency of cRGD ranged between 40–70% whereas conjugation efficiency of cysteine was approximately 40%. Theoretically, 100% conjugation efficiency could have been reached because a molar excess of cysteine to maleimide was used in the reaction. Nevertheless, previous work conducted in our group has indicated that the surface availability of maleimide groups might actually be lower than expected [45], probably due to the miscibility of PEG and PLGA [49,50] that leads to the solubilization of the maleimide groups in the polymer matrix of the NPs and thus their inaccessibility for reaction with thiolyated cRGD. Cytotoxicity of PLGA-based NPs, incubated with HUVECs and other cell types, was reported by us and other groups to be negligible [51,52,53,54], and the coupling of cRGD to the surface of PLGA NPs is not expected to cause cytotoxicity as discussed by Graf et al. [23].

Upon static incubation with HUVECs, cell association was virtually absent for Cys-NPs (non-targeted control). On the contrary, cell association was observed for cRGD-NPs with all three ligand densities, and the fluorescent signal of the NPs was distributed homogeneously over the cell population. The association of cRGD-NPs to HUVECs showed a positive correlation with the NPs concentration, the incubation time and the cRGD surface density. In addition, confocal microscopy analysis was used to gain insight into the cellular localization of the fluorescent signal detected by flow cytometry and thus on the intracellular distribution of the NPs.

The much higher cell association and uptake of cRGD-NPs in comparison to Cys-NPs suggests that cRGD mediates the binding of the NPs to HUVECs. Other studies comparing non-targeted and targeted PEGylated PLGA NPs (~100–200 nm) also found increased binding to and uptake by HUVECs for targeted NPs [22,55]. Nevertheless, non-specific cell interactions of non-targeted NPs were also observed, particularly at incubation times ≥ 1 h. These findings are in contrast with the low cell association observed for the Cys-NPs in our study even after 3 h of incubation. Endocytosis of non-targeted NPs is highly influenced, amongst others, by their size, charge and surface chemistry. The Cys-NPs used in this study have a relatively large size (~300 nm) which would limit their caveolae-mediated uptake and their uptake by other non-clathrin mediated mechanisms generally associated with the formation of small vesicles (~50–100 nm) [56,57]. Additionally, Cys-NPs have a slightly negative surface charge (zeta potential ~−12 mV), which limits the charge-driven NPs-cell interaction and uptake that is often reported for positively charged NPs [58,59]. Particles decorated with RGD can be internalized by receptor-mediated endocytosis [60] or by phagocytosis [61] driven by the interaction between RGD and α_v_β_3_. In the present study, the association of cRGD-NPs to HUVECs was time- and concentration-dependent, which is in line with reports on the interactions between other RGD-functionalized nanocarriers and α_v_β_3_ overexpressing cells [62,63]. Cell association of NPs was also highly influenced by the surface density of cRGD. Similarly, other research groups have found that increasing RGD-densities on the surface of nanocarriers generally result in increasing binding to the integrin α_v_β_3_, coated on plate surfaces or present on the surface of cells [64,65]. In the present work, the NPs association to HUVECs increased with increasing RGD-densities and reached a maximum at ~118,000 RGD/µm^2^ (density on the high cRGD-NPs). In a similar study with PLGA-NPs prepared with cRGD densities ~26,000–137,000 RGD/µm^2^, the association to HUVECs was also dependent on the cRGD density and also reached a maximum for the NPs with the highest surface density [66]. However, this study was only performed under static conditions. The increased cell association observed for increasing cRGD densities on the NPs can be attributed to the multivalent interactions between this peptide and the integrin α_v_β_3_. In line with our findings, several authors have reported that multivalency in RGD systems enhances α_v_β_3_ integrin binding and internalization [67,68,69,70,71].

In (patho)physiological conditions, the contact between cRGD-NPs and endothelial cells in (angiogenic) blood vessels will be subjected to the effects of constant blood flow (i.e., shear stress and presence of blood cells). These effects probably modify the way the NPs interact with the cells compared to static conditions frequently used in in vitro assays that aim to study cellular interactions. For this reason, nanoparticle-cell interactions were studied under flow in media of different complexities (EBM-2 cell culture medium and washed RBCs) and at different incubation times (1 and 16 h). These experiments were conducted at a shear rate of 300 s^−1^, which is comparable to the mean shear rate reported for the human carotid artery [72,73]. Unfortunately, a direct comparison between cell association in static and flow conditions is not possible, since this was determined using different methods for each condition, i.e., flow cytometry for static and image analysis for flow.

The striking increment in cell association observed for all cRGD-NPs in medium containing RBCs can be partly attributed to the increased shear stress in this medium (at equal shear rates). In fact, shear stress-related differences in cellular binding/uptake of NPs have been previously reported. While some have shown an inverse correlation between uptake and shear rates [35,37,74], others showed a direct correlation between these parameters [75]. Differences in the composition and physicochemical characteristics of the nanocarriers evaluated, as well as in the cell types and experimental setups, could contribute to the discrepancies between these studies. The differences in association of cRGD-NPs to HUVECs observed under flow with and without washed RBCs, indicate that these blood components influence the association of the NPs with the endothelial cells. It has been proposed that the presence of RBCs facilitates the contact of nanocarriers with vascular endothelium by two main mechanisms: (1) RBCs localize mainly to the center of blood vessels, pushing smaller particles close to the vessel wall [76,77] and (2) RBCs rotate and tumble, colliding with small particles and pushing them towards the vessel wall [76,78,79]. The presence of RBCs in the medium has indeed been shown to enhance the binding of adhesion molecule-targeted particles to HUVECs under flow conditions [80,81]. Though these studies were carried out using particles with relatively large sizes (0.5–10 µm). Our findings, as well as previously published data [78], indicate that RBCs also have an effect on the cell association of smaller size NPs (~300 nm). In future studies, more insights into the behavior of different types of NPs (i.e., size, charge, composition, etc.) in conditions approximating physiological settings could be gained by conducting flow experiments in the presence of whole blood. Additionally, this perfusion set-up can also be employed to study the delivery of cRGD-PLGA particles loaded with therapeutic compounds such as anti-angiogenic and pro-angiogenic drugs.

## 5. Conclusions

The results presented in this work show that polymeric NPs functionalized with cRGD can target αvβ3 integrins expressed by endothelial cells, and that cell association of these nanocarriers is positively correlated with the surface density of cRGD. Targeting of endothelial cells was evaluated using two approaches, under static and flow conditions. Experiments conducted using washed RBCs in the medium, which represent more realistic and physiologically relevant conditions, showed that the presence of these blood components facilitates the association of NPs to endothelial cells under flow. The setup used for the experiments under flow is, as such, particularly appropriate for the study of endothelial cell targeting, rather than tumor cell targeting. This approach is likely a step forward in predicting the in vivo association of endothelial-cell targeted nanocarriers administered intravenously. Such an in vitro assay would therefore allow for high-throughput screening of different targeted formulations and is a potential asset for the reduction of laboratory animal testing.

## Figures and Tables

**Figure 1 nanomaterials-10-01353-f001:**
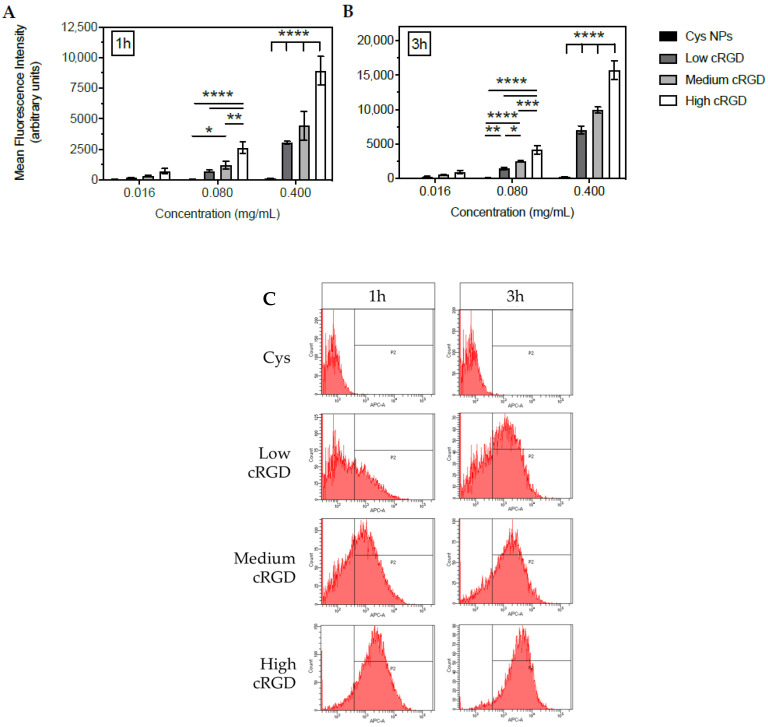
Association of Cys-NPs or cyclic Arg-Gly-Asp (cRGD)-NPs of different ligand densities to human umbilical vein endothelial cells (HUVECs) under static conditions at 37 °C. (**A**) Association measured by flow cytometry after 1 h, and (**B**) 3 h of incubation (n = 4). (**C**) Representative histograms of the fluorescence distribution in HUVECs incubated with 0.08 mg/mL of Cys-NPs or cRGD-NPs for 1 or 3 h. Data (mean ± SD) were analyzed by one-way ANOVA followed by Tukey’s multiple comparison test; * *p*-value < 0.05, ** *p*-value < 0.01, *** *p*-value < 0.001 and **** *p*-value < 0.0001.

**Figure 2 nanomaterials-10-01353-f002:**
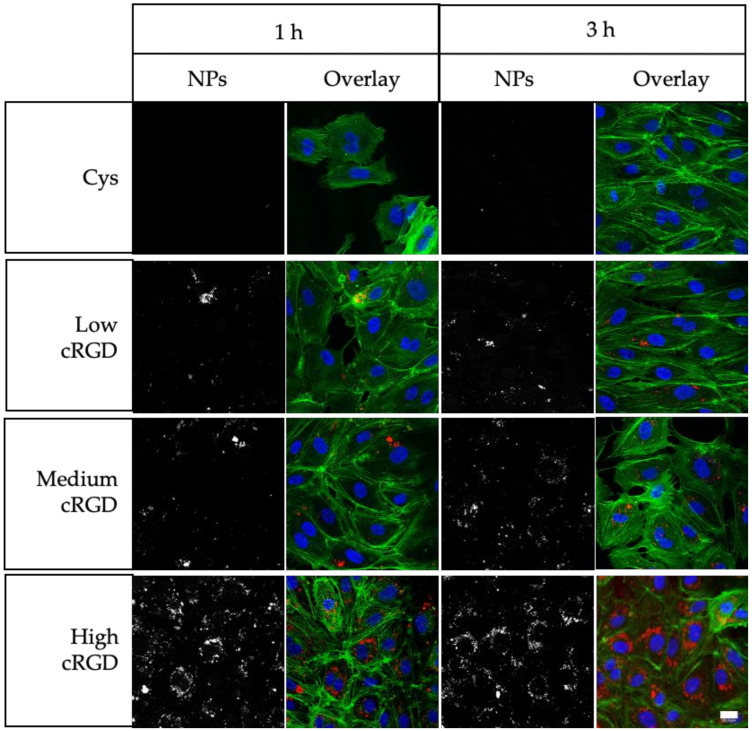
Uptake of Cys-NPs and cRGD-NPs of different ligand densities by HUVECs under static incubation conditions. Confocal cross-sections of HUVECs incubated with NPs for 1 and 3 h at 37 °C. Nuclei are stained in blue, cytoskeleton (F-actin) is stained in green and NPs are depicted in red. Scale bar represents 20 μm.

**Figure 3 nanomaterials-10-01353-f003:**
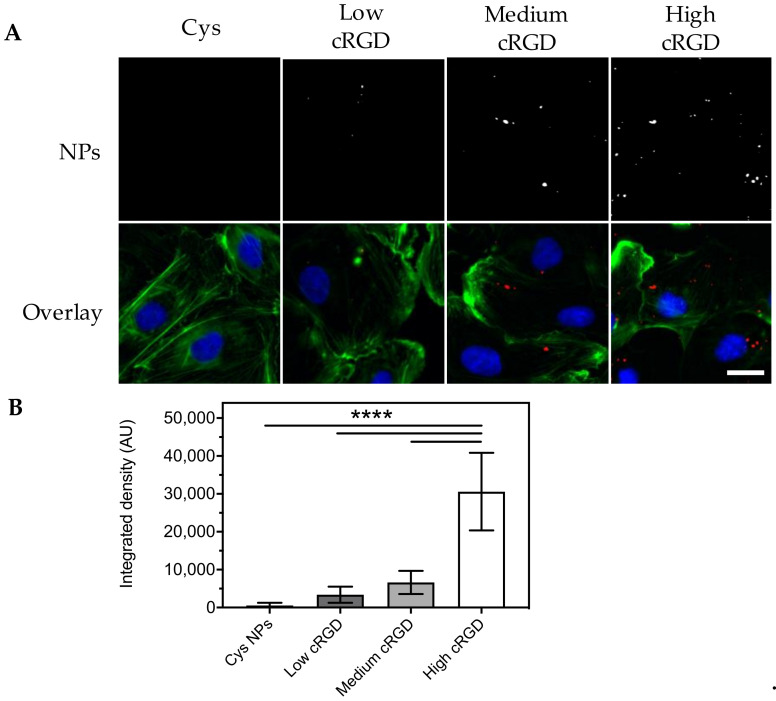
Association of Cys-NPs and cRGD-NPs of different ligand densities to HUVECs after 1 h incubation under flow (shear rate 300 s^−1^) in Endothelial Basal Medium (EBM-2) at 37 °C. (**A**) Representative images obtained with an epifluorescence microscope (cropped images). Nuclei are stained in blue, cytoskeleton (F-actin) is stained in green and NPs are observed in red. The same brightness and contrast settings were used for the red channel in all images. (**B**) Image analysis: fluorescent signal for the NPs is reported as integrated density (n = 10). All images obtained from the different formulations were processed in the same manner (background subtraction and equal thresholding). Data (mean ± SD) were analyzed by one-way ANOVA followed by Tukey’s multiple comparison test; **** *p*-value < 0.0001. Scale bar represents 20 μm.

**Figure 4 nanomaterials-10-01353-f004:**
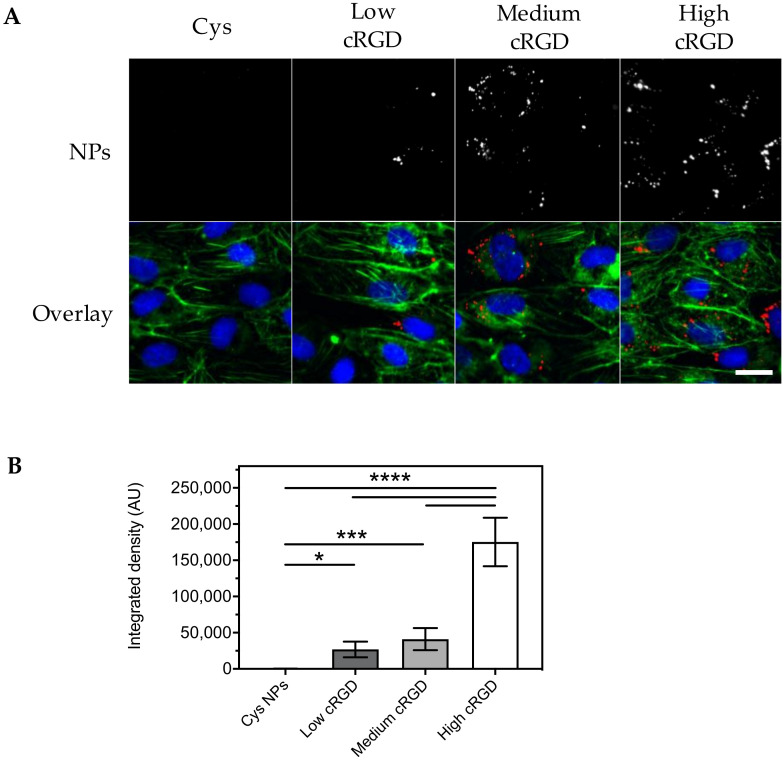
Association of Cys-NPs and cRGD-NPs of different ligand densities to HUVECs after 16 h incubation under flow (shear rate 300 s^−1^) in EBM-2 medium at 37 °C. (**A**) Representative images obtained with an epifluorescence microscope (cropped images). Nuclei are stained in blue, cytoskeleton (F-actin) is stained in green and NPs are observed in red. The same brightness and contrast settings were used for the red channel in all images. (**B**) Image analysis: fluorescent signal for the NPs is reported as integrated density (n = 10). All images obtained from the different formulations were processed in the same manner (background subtraction and equal thresholding). Data (mean ± SD) were analyzed by one-way ANOVA followed by Tukey’s multiple comparison test; * *p*-value < 0.05, *** *p*-value < 0.001 and **** *p*-value < 0.0001. Scale bar represents 20 μm.

**Figure 5 nanomaterials-10-01353-f005:**
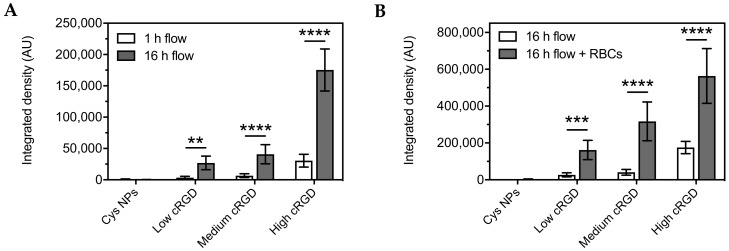
Comparison of the cell association of Cys-NPs and cRGD-NPs of different ligand densities to HUVECs at 37 °C under flow (shear rate 300 s^−1^) at different conditions. (**A**) Cell association after 1 or 16 h of incubation in EBM-2 medium. (**B**) Cell association after 16 h of incubation in EBM-2 medium with and without washed RBCs. The fluorescent signal for the NPs is reported as integrated density (n = 10). Data (mean ± SD) were analyzed by two-way ANOVA followed by Sidak’s multiple comparison test; ** *p*-value < 0.01, *** *p*-value < 0.001 and **** *p*-value < 0.0001.

**Figure 6 nanomaterials-10-01353-f006:**
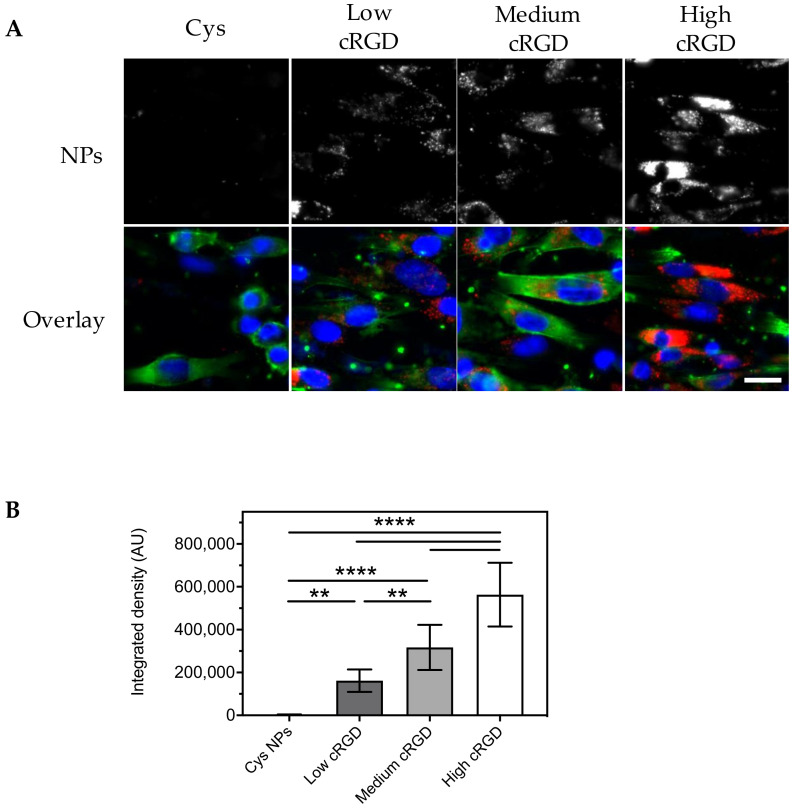
Association of Cys-NPs and cRGD-NPs of different ligand densities to HUVECs after 16 h incubation under flow (shear rate 300 s^−1^) in EBM-2 medium containing washed red blood cells (RBCs) (hematocrit 32%) at 37 °C. (**A**) Representative images obtained with an epifluorescence microscope (cropped images). Nuclei are stained in blue, cytoskeleton (F-actin) is stained in green and NPs are observed in red. The same brightness and contrast settings were used for the red channel in all images. (**B**) Image analysis: fluorescent signal for the NPs is reported as integrated density (n = 10). All images obtained from the different formulations were processed in the same manner (background subtraction and equal thresholding). Data (mean ± SD) were analyzed by one-way ANOVA followed by Tukey’s multiple comparison test; ** *p*-value < 0.01, and **** *p*-value < 0.0001. Scale bar represents 20 μm.

**Table 1 nanomaterials-10-01353-t001:** Z-average diameter, polydispersity index, zeta potential and conjugation efficiency of Cy5-poly(lactic-co-glycolic acid) (PLGA) nanoparticles (NPs).

Formulation	Diameter (nm)	PDI	Zeta Potential (mV)	Conjugation Efficiency (%) *^a^*	cRGD Peptides Coupled per NP *^b^*	Density (cRGD/ µm^2^) *^b^*
**NPs**	303 ± 34	0.25 ± 0.10	−10.1 ± 0.7	NA *^c^*	NA *^c^*	NA *^c^*
**Cys-NPs**	316 ± 19	0.22 ± 0.02	−12.2 ± 0.6	39 ± 12	NA *^c^*	NA *^c^*
**Low cRGD-NPs**	313 ± 11	0.21 ± 0.00	−12.0 ± 0.4	38 ± 5	~3400	~11,000
**Medium cRGD-NPs**	312 ± 14	0.19 ± 0.03	−11.4 ± 0.2	50 ± 8	~9700	~31,700
**High cRGD-NPs**	332 ± 22	0.23 ± 0.04	−10.1 ± 0.6	71 ± 8	~40,700	~117,600

Independently prepared batches n = 4. *^a^* Conjugation efficiency = (1–([ligand in the supernatant]/[ligand added in the conjugation reaction])) × 100%), *^b^* Calculated as reported in Appendix A, *^c^* Not applicable. All data are mean ± SD.

**Table 2 nanomaterials-10-01353-t002:** Fluorescence distribution (% of positive cells) in HUVECs upon static incubation at 37 °C with Cys-NPs and cRGD-NPs of different ligand densities.

Concentration of Fluorescent NPs (mg/mL)	Percentage of Fluorescence Positive HUVECs							
	Cys-NPs		Low cRGD-NPs		Med cRGD-NPs		High cRGD-NPs	
	1 h	3 h	1 h	3 h	1 h	3 h	1 h	3 h
0.016	0.3 ± 0.2	0.2 ± 0.1	8 ± 1	15 ± 5	15 ± 4	36 ± 3	43 ± 8	60 ± 9
0.080	0.5 ± 0.2	0.9 ± 0.4	30 ± 4	58 ± 8	51 ± 11	83 ± 2	83 ± 5	93 ± 3
0.400	3 ± 1	12 ± 7	71 ± 1	91 ± 4	85 ± 5	98 ± 1	97 ± 1	99 ± 0

Percentage of fluorescence positive HUVECs (P2) obtained from a gated HUVECs population (P1), n = 4. All data are mean ± SD.

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
