# Peer review of "Endothelial Cell Targeting by cRGD-Functionalized Polymeric Nanoparticles under Static and Flow Conditions"

_nanomaterials, 2020, doi:10.3390/nano10071353_

Round 1

Reviewer 1 Report

The authors have evaluated the impact of various NPs (targeted & non-targeted) on endothelial cells under static and flow conditions to provide insights on targeted therapy.

There few concerns that the authors need to address:

  1. Authors have used different concentrations of NPs for static (0.8mg/ml) and flow (0.04mg/ml) conditions. It is understandable that due to flow conditions, the overall exposure of the ECs to the NPs would be much higher under flow conditions esp with the 16hr exposure conditions. Authors to include in the methods/ discussion the rational for chosing different concentrations for static and flow conditions. Also to include the rationale for choosing 0.04mg/ml for flow conditions. Further, to include the overall exposure of ECs to the NPs under 3hr and 16hr flow conditions (which should be quantifiable based on flow rate).
  2. It is known that NPs perse could have cytotoxicity potential. Authors to include viability data under static & flow (3hr & 16hr) conditions.
  3. For the localization studies using confocal microscopy, authors have used Phalloidin to demonstrate the intracellular localization. Since authors have used confocal microscopy to image the cells, authors to include cross-section or z-profiles. Cross-sectional profiles would help to visualize if the NPs are present within the cells or on-top of the cells.
  4. For all the results, authors to include statistical analysis.
  5. Lastly, it is not clear on the future aspects of the findings. Is the findings of this study targetted towards anti-angiogenic or angiogenesis-related therapies!
  6. Since, Integrin αvβ3 is associated with angiogenesis, it would add value to investigate the impact of the targetted NPs on angiogenesis. Authors can include this data or in the discussion.

Reviewer 2 Report

In this submission, Martínez-Jothar et al. study the effect of cRGD ligand density, concentration, incubation time, and flow conditions on the targeting of nanoparticles to vascular endothelial cells. Efficient targeting of angiogenic endothelium is a longstanding goal in nanomedicine, and systematic studies that characterize structure-function phenomena would be of broad interest to the field. The particle synthesis and characterization work presented here is reasonable and well-described. The manuscript is clear and contains sufficiently-detailed methods. However, the cell studies are somewhat superficial and lack critical analyses. Additional information would be needed for the reader to evaluate the context and strength of the results presented. 

1) The submission lacks statistical analysis - the quantitative data presented should be analyzed using appropriate statistical tests. The variability described in the tables and error bars shown in the figures should be defined (SD or SEM?).

2) Since the authors chose few time and concentration points, some justification should be given for the selected conditions. Is the MFI vs. concentration trend expected to be linear as depicted? Does the increase in MFI plateau after 3 hours?

3) In the fluorescence images meant to show intracellular accumulation of fluorescent nanoparticles, the authors claim that the punctate nature of the signal indicates endosomal localization. This needs to be supported by co-localization with an endolysosomal marker or similar. Furthermore, the confocal microscopy conditions are not described. Are the images a maximum projection or slices? As shown, it cannot be determined whether the signal is derived from intracellular or surface-bound nanoparticles. A cross-section projection would clear up this ambiguity. All of the figures with microscopy images require a scale bar. 

4) Cell viability studies are critical to give context to the authors’ results. An alternative hypothesis could be that the cells exposed to higher concentrations of nanoparticles or flow conditions become more permeable due to stress or disruption of the monolayer. The data would be unreliable if the cells are apoptotic. The gross morphology of the cells appears different between conditions, and the cropped images are too small to evaluate the integrity of the monolayer. It must be shown that the experimental conditions do not fundamentally change the nature of the target cells. 

Minor comments:

1) The authors should comment on the differences seen in conjugation efficiency between low, medium, and high cRGD densities. If steric hindrance/maleimide availability is the cause, the trend would be opposite of that is reported. 

2) If it is known where αvβ3 integrins are expressed on vascular cells (basal vs. apical), this should be mentioned. The cells used for flow experiments are cultured for two days longer before nanoparticle exposure than those used in static experiments. Would this be expected to affect tight junction establishment or integrin expression?

3) Line 305: -10-12 mV is ambiguous 

4) Lines 213, 343, 344: FACS should read flow cytometry unless cells were sorted

5) Figure 1: Inconsistent with RGD and cRGD

Reviewer 3 Report

Major comments

The authors have synthesized Cy5 labelled maleimide-PEG-PLGA nanoparticles with integrated cRGD peptides at different concentration (low, medium, high). The addition of NPs to endothelial cells (HUVECs) under static conditions showed that NPs can integrate into HUVEC’s membranes. NPs not loaded with cRGD peptides failed to associate with HUVECs. Similarly, cRGD-NPs associated with HUVECs under flow conditions.

The authors’ main conclusion is that “polymeric NPs functionalized with cRGD can target endothelial cells”. This is not supported by data, as authors have not tested the ability of cRGD-NPs to be associated with other cells (e.g. Immune cells such as PBMCs). Thus, the fact that cRGD-NPs can associate with HUVECs does not mean cRGD-NPs target HUVECs. Another conclusion of this study is that cRGD higher density promotes NP association with HUVECs. This finding is hardly surprising and seems self-evident.  Finally, the authors showed that cRGD-NPs can associate with HUVECs under both static (in culture) or flow (physiological scenario) conditions.

Despite the manuscript is not short, the study results are too preliminary. The study is lacking hypothesis, and it is unclear  why maleimide-PEG-PLGA NPs were used in this study. The reason to study maleimide-PEG-PLGA NPs binding to HUVECs is puzzling too. What is the purpose of targeting endothelial cells?

Minor comments

Immunofluorescent images are lacking error bars

Round 2

Reviewer 1 Report

Authors have well addressed the queries raised.

Author Response

We would like to thank the reviewer for his/her comment.

Reviewer 2 Report

The authors have improved the manuscript somewhat with the changes included in the revision. The type of error included with the values presented in Tables 1&2 is still not defined. This should be corrected prior to publication.

Author Response

We would like to thank the reviewer for this comment. We have added the requested information to Table 1 and Table 2.